# Methodology and Experimental Protocol for Studying Learning and Motor Control in Neuromuscular Structures in Pilates

**DOI:** 10.3390/healthcare12020229

**Published:** 2024-01-17

**Authors:** Mário José Pereira, Alexandra André, Mário Monteiro, Maria António Castro, Rui Mendes, Fernando Martins, Ricardo Gomes, Vasco Vaz, Gonçalo Dias

**Affiliations:** 1Faculty of Sports Sciences and Physical Education, University of Coimbra, 3000-214 Coimbra, Portugal; vascovaz@fcdef.uc.pt; 2Coimbra Health School, Polytechnic Institute of Coimbra, 3046-854 Coimbra, Portugal; alexandra.andre@estescoimbra.pt (A.A.); mmonteiro@estesc.ipc.pt (M.M.); 3Laboratory IIA, ROBOCORP, Polytechnic Institute of Coimbra, 3045-093 Coimbra, Portugal; maria.castro@ipleiria.pt (M.A.C.); rmendes@esec.pt (R.M.); fmlmartins@esec.pt (F.M.); rimgomes@esec.pt (R.G.); goncalodias@esec.pt (G.D.); 4School of Health Sciences, Polytechnic Institute of Leiria, 2411-901 Leiria, Portugal; 5Centre of Mechanical Engineering, Materials and Processes (CEMMPRE), University of Coimbra, 3000-214 Coimbra, Portugal; 6ESEC-UNICID-ASSERT, Polytechnic Institute of Coimbra, 3045-093 Coimbra, Portugal; 7CIDAF (lida/dtp/04213/2020), University of Coimbra, 3000-214 Coimbra, Portugal; 8Coimbra Education School, Polytechnic Institute of Coimbra, 3030-329 Coimbra, Portugal; 9Instituto de Telecomunicações (IT), Delegação da Covilhã, 6201-001 Covilhã, Portugal; 10InED—Centre for Research and Innovation in Education, Porto Polytechnic Institute, 4200-465 Porto, Portugal

**Keywords:** pilates, electroencephalography, gaze behavior, abdominal ultrasound, elastography, motor control

## Abstract

The benefits of Pilates have been extensively researched for their impact on muscular, psychological, and cardiac health, as well as body composition, among other aspects. This study aims to investigate the influence of the Pilates method on the learning process, motor control, and neuromuscular trunk stabilization, specifically in both experienced and inexperienced practitioners. This semi-randomized controlled trial compares the level of experience among 36 Pilates practitioners in terms of motor control and learning of two Pilates-based skills: standing plank and side crisscross. Data will be collected using various assessment methods, including abdominal wall muscle ultrasound (AWMUS), shear wave elastography (SWE), gaze behavior (GA) assessment, electroencephalography (EEG), and video motion. Significant intra- and inter-individual variations are expected, due to the diverse morphological and psychomotor profiles in the sample. The adoption of both linear and non-linear analyses will provide a comprehensive evaluation of how neuromuscular structures evolve over time and space, offering both quantitative and qualitative insights. Non-linear analysis is expected to reveal higher entropy in the expert group compared to non-experts, signifying greater complexity in their motor control. In terms of stability, experts are likely to exhibit higher Lyapunov exponent values, indicating enhanced stability and coordination, along with lower Hurst exponent values. In elastography, experienced practitioners are expected to display higher transversus abdominis (TrA) muscle elasticity, due to their proficiency. Concerning GA, non-experts are expected to demonstrate more saccades, focus on more Areas of Interest (AOIs), and shorter fixation times, as experts are presumed to have more efficient gaze control. In EEG, we anticipate higher theta wave values in the non-expert group compared to the expert group. These expectations draw from similar studies in elastography and correlated research in eye tracking and EEG. They are consistent with the principles of the Pilates Method and other scientific knowledge in related techniques.

## 1. Introduction

The Pilates Method was developed by Joseph Hubertus Pilates in the early 1920s. It aims to holistically improve physical conditioning by focusing on the dualism of the body and mind. This technique targets trunk stability, strength, flexibility, posture, and breathing. The method is based on six principles: center, concentration, control, precision, fluidity, and breathing. It can be practiced alone or in groups, with or without equipment such as a reformer, Cadillac, arcs, Swiss balls, or bands [1,2].

The popularity of Pilates has led to an increase in research on its potential benefits [3]. However, the current limitations include (i) great sample heterogeneity; (ii) predominance of female sex; (iii) absence of effective assessment of long-term effects; (iv) variation in types of intervention (e.g., with or without equipment); and (v) the study of the cost–benefit ratio [4]. To address these limitations, we propose a multifactorial analysis approach based on dynamic, ecological, and cognitive perspectives [5,6]. Our study incorporates echography, electroencephalography, gaze behavior, and kinematic data to provide a broader understanding of the effects and potential benefits of Pilates in the current context [7].

Research on Pilates has primarily focused on its effects and benefits on core muscles [8]. With the large number of techniques and muscles assessed with ultrasound imaging echography and electromyography, the results have varied between the positive effects of this method and the absence of influence on core stabilization muscles [9,10,11]. In this matter, our study focuses on the following set of four muscles: rectus abdominis (RA), obliquus internus abdominis (IO), oblique externus abdominis (EO) and transversus abdominis (TrA). Despite some debate around the inclusion of muscles within the “core”, our chosen combination incorporates specific muscle groups and the diaphragm as they play a crucial role in forming a framework essential for generating intra-abdominal tension. This tension not only helps to stabilize the spine but also holds great significance in terms of human movement and displacement [12]. As muscle elasticity has a strong association with muscle tension, in our study we will add elastography (SWE) to 2D ultrasound (ECG) in an attempt to introduce new data with more precision regarding the advantages of Pilates and trunk stability [13].

The influence of Pilates practice on neuro-muscular and cognitive function is interesting to address. Thus, research on physical activity using EEG signals, combined with nonlinear methods, will lead to new horizons in brain function research [14,15]. Recent studies point to advantages to cognitive performance in athletes compared to non-athletes [16], and it will be pertinent to obtain this knowledge in other types of activities and physical activity levels. In fact, new technologies and devices can use EEGs to study simple movements, introducing the possibility of investigating new experimental designs [17,18,19].

The gaze behavior associated with motor imagery obtained from mirror images may provide information regarding the motor control, learning speed and movement quality [20,21,22]. In activities with precise movements, such as dance, Pilates, or martial arts, the impact of the presence of a mirror and the gaze behavior of the practitioners still needs research [23]. Furthermore, the output of quality of movement, particularly if associated with other data, could be very useful for obtaining a clearer glimpse of these phenomena [24].

The objective of this study is to investigate how the Pilates method affects the learning process, motor control, and neuromuscular stabilization of the trunk, specifically in experienced and inexperienced practitioners. Secondly, it aims to identify differences in strength and muscle recruitment among the four selected muscles that we are studying. Finally, we aim to build a pattern or map of brain waves and gaze behavior, which will be created according to the practitioner’s level of experience and length of practice.

## 2. Materials and Methods

### 2.1. Study Settings and Design

This semi-randomized controlled trial will compare the level of experience in various aspects of motor control and learning in two Pilates-based skills (standing plank, and the side crisscross. See Appendix A—Figure A1). The study protocol has been approved by the Ethics Committee of the Faculty of Sport and Physical Education at University of Coimbra (number CE/FCDEF-UC/00042022) and was submitted and approved by their Scientific Council. Table 1 summarizes the combination of variables and outcomes that are relevant to our study goals.

The experience level of the participants is an independent variable manipulated through the presence or absence of a mirror during the exercises [25]. This allows control of possible variables that could impact the object of the study. Both Pilates-based exercises also provide a means of controlling for these variables and ensuring that the exercises are unfamiliar to all participants. Environmental factors and organic characteristics of the participants will not be used as they are not relevant to the objective of the study. However, Body Mass Index (BMI) can be considered due to its correlation with strength [26]. These design choices result in suitable statistical analyses using linear or nonlinear methods.

### 2.2. Sample

We will assemble a wide-ranging and reliable sample, using a semi-randomized controlled experimental study, ensuring that our methodology is implemented. For the experimental study to produce reliable results, 39 participants will be involved, each with specific selected characteristics (Table 2). To ensure we have enough participants for our study, we used G*Power (version 3.1.9.6—software designed by the University of Kiel, Germany). Our study required a two-tailed test, with an alpha level of 0.05, power of 90%, and an effect size (d) of 1.39, which were taken into consideration during the design of our study [27,28].

#### Eligibility Criteria

The sample for the experienced group will be selected from a local Pilates studio. Participants must have more than six months of practice, with more than 1 h of practice per week [27]. The non-expert group will be composed of subjects who must not have had any Pilates practice in the last three months. For the control group, in addition to not practicing any form of Pilates in the last three months, we also require that they not practice any form of Pilates during the study. These subjects will be allowed to maintain their physical activity routines or create new routines if body and mind activities are not included (yoga, body balance, tai-chi, among others). All groups will be composed of 13 participants between 30 and 50 years of age. The exclusion criteria are pregnancy and medical contraindications to practicing physical exercise [29].

### 2.3. Intervention

The expert group will maintain their normal Pilates activity (at least once per week/60 min) in their usual facilities. The only addition to their regular routines will be the two Pilates skills (standing plank and the side crisscross) introduced and evaluated in our studies. All instructors involved in these facilities will have specific training to be able to teach and supervise these two new skills (four hours).

These two exercises will be performed in a type of sliding machine called a Standing Machine; participants will be able to perform the exercises in various positions, adding some instability, but particularly in orthostasis. In the plank, participants maintain a straight back and then flex and extend the feet for eight cycles (Appendix C—Table A2). In the side crisscross exercise, the participant maintains a lateral position while a crossing of limbs is done repeatedly for fifteen cycles (Appendix C—Table A3). The Standing Machine is patented pending number No 118908 in National Institute of Industrial Property.

All instructors will have POLESTAR^®^ Pilates training completed between 2011 and 2015. The non-expert group will start a weekly program in one of the Pilates studios participating in this study. This group has a weekly standardized session (55 min average) according to Appendix B—Table A1. These participants perform two laps around the circuit, performing 8 to 10 repetitions in each station, performing the standing plank on the first lap and the side crisscross on the second.

### 2.4. Outcomes

The outcomes associated with this project are relative to each study. They are briefly described below, despite being subject to detailed methodological protocols in each study.

#### 2.4.1. Abdominal Wall Muscle Ultrasound (AWMUS)

The ultrasound study will collect data regarding RA, IO, EO, and TrA thicknesses [30]. Based on the anatomical assessment markers for each muscle, as presented in Table 3, data will be collected on the right side of the participants’ trunks both at the initial assessment and after twelve weeks of intervention. Participants lay in the supine position with their arms extended along their torso. Data will be collected in millimeters after 3 measures on the left side of the body between superficial and deeper borders of muscles to ensure the quality of measurements [31]. The mean values will be calculated to obtain a final value for each muscle in each condition. Images of the EO, IO, and TrA muscles were obtained with the transducer positioned vertically to the participant’s trunk, ten centimeters to the left of the navel, at the midpoint between the iliac crest and the last rib [32]. In the RA muscle, the transducer will be positioned horizontally right above the navel [31]. A portable ultrasound (Acuson P500) with a transducer (Siemens Healthcare GmbH, Erlangen, Germany) will be operated by one experienced researcher (>10 years). The collections will be made (a) during rest and while performing the abdominal hollowing maneuver (AHM), and (b) during a standing plank with arms back at the end of 8 repetitions.

#### 2.4.2. Shear Wave Elastography (SWE) Assessment

The SWE elastography will collect complementary additional and improved data. In fact, the muscles analyzed are fundamental for trunk stability. Their stiffness and the consequent intra-abdominal tension are important in health and motion [12,33,34]. Shear-wave elastography (SWE) data collection uses the muscle shear module using an Acuson Sequoia Ultrasound System 2018 (Siemens Healthcare GmbH, Erlangen, Germany). This device is coupled with a linear transducer array (SL10-4, 4–10 MHz, Siemens Healthcare GmbH, Erlangen, Germany) in the shear-wave elastography mode, namely the musculoskeletal pre-set and B-mode [35].

The shear-wave elastography system uses the mechanical properties of muscle hardness and represents the transverse’s muscle stiffness. In technical terms, shear waves are generated within the muscle under evaluation, propagating through it. These waves are then quantified by measuring the shear wave velocity (Vs) using a specific algorithm, all in a non-invasive manner (see Figure 1). A higher shear wave velocity indicates greater muscle elasticity. This method is applicable even for assessing muscles at deeper layers. The shear modulus (μ) is calculated using the shear wave velocity (Vs) as follows: μ = ρ × Vs^2^, where ρ represents the muscle mass density (1000 kg/m^3^) [36]. The push frequency is automatically configured by the ultrasound equipment to approximately 1 Hz, falling within the range of 0.8 to 1.4 Hz [13,37].

The procedure will be carried out by the same experienced ultrasound operator, to avoid interobserver variation. The ultrasound device, transducer, setup parameters, and assessment locations will be identical for all assessments. The transducer location for the TrA muscle was established previously. The determination of the transducer location for all muscles is based on methodologies employed in prior SWE studies (refer to Table 3). The examiner will use a waterproof ink pen to mark the target region directly on the subject’s skin. This target region is situated on the right side of the umbilicus and extends 2 cm inward from the mid-axillary line. The linear transducer array is positioned on the target region in a transverse orientation to the body’s long axis, aligning it parallel to the muscle fibers of the TrA. The apparatus measures shear wave speed and calculates elasticity using the provided equation [13].

The assessments are made with the subjects in a supine position at rest and using the abdominal hollowing maneuver. Three shear modulus data points will be recorded for each muscle where the average mean is used as the relative value [38]. The system will record the shear wave velocity in meters per second (m/s), the median elasticity in kilopascals (kPa), and the measurements for depth and diameter in centimeters. Additionally, the SWE results can be presented in the form of an elasticity map, where different colors are assigned to represent varying values, creating a heatmap that visualizes the data.

#### 2.4.3. Body Composition Analysis

Dual-energy X-ray absorptiometry (DEXA) has been widely used in body composition (BC) assessments. This process involves the utilization of two previously known X-ray energies. Subsequently, the constituent photons of the X-ray beam interact with the atoms within the subject’s body, resulting in a reduction of beam intensity, a phenomenon known as beam attenuation. Finally, this attenuated beam interacts with a detector [39,40]. This process, known as attenuation, is related to the absorption, or scattering of photons depending on the thickness, density, and atomic composition of the cross structures. As so, low-density tissues attenuate the beam less than tissues with higher density, like bone tissue. Each tissue has a different attenuation and allows the DEXA the capacity to measure the coefficient of attenuation of the two power spikes issued [41]. DEXA provides BC estimations dividing the body into lean mass (LM), fat mass (FM) and bone mineral content.

For the assessment of BC with DEXA, the Biphoton Bone Densitometry Lunar—GE Healthcare, Lunar iDXA model, Software Lunar Encore for Windows version, 13.60.300 (Waltham, MA, USA) will be used (Figure 2).

The radiation used in DEXA procedures is extremely low, at 1.8/2.1 µSv for adults of both sexes [42]. This method has been used in various sports and activities such as football [43,44], rugby [45], swimming [46], and others related to physical fitness [26].

The initial step involves daily calibration in accordance with the manufacturer’s instructions. Participants wear light clothing, gym shorts and a sleeveless shirt, with no high-density artefacts on the body or clothing (metallic, plastic). The subjects are positioned in a dorsal decubitus posture within the designated evaluation area (rectangle), aligning their spine with the central longitudinal axis of the table.

Body symmetry will be confirmed and ensured. The lower limbs are fully extended, while the upper limbs were likewise extended and placed alongside the trunk, maintaining a separation from the trunk to facilitate individualized processing. The palms of the hands rest on the table in a pronated position. Total immobility is advised and confirmed for all individuals, and they are requested to refrain from talking during the assessment. All procedures, data collection, and analysis will be carried out by the same operators with more than 10 years of experience.

The outcome report includes all body composition (BC) parameters expressed in absolute values in grams or percentages, following the methodology [39]. Furthermore, dual-energy X-ray absorptiometry (DEXA) provides insights into the partial distribution of lean mass (LM), fat mass (FM), and bone mineral density (BMD) [47,48].

#### 2.4.4. Gaze Behavior Assessment

Eye tracking technology allows the study of the participants’ ocular behavior, i.e., the analysis of their eye movements and gaze behaviors during Pilates. This allows the study of smooth pursuit behaviors, saccades patterns, and fixation points, contributing to a greater understanding of the gaze behavior of Pilates participants. Upon reaching the “calibration area”, the subjects are required to focus on the calibration card, which is essential for configuring the TOBII Pro Glasses 3. This calibration procedure is repeated when the subjects switch sides (for bilateral exercises) and when they change exercises. Subsequently, the collected data will be processed using TOBII PRO LAB, following the predefined dynamic Area of Interest (AOI). Following this, the analysis will encompass time and frequency of saccades, fixations, and smooth pursuit behaviors. These AOIs are associated with the quality of movement and the success of movement execution, depending on the specific exercise being performed, with a focus on the limbs and/or hips [49]. In the analysis of gaze behavior, the collected data should be correlated with the number of saccades and the duration of each fixation within the predefined Area of Interest (AOI), tailored to each Pilates exercise. This information can also be visually represented as a heatmap, providing a spatial and temporal visualization of gaze patterns during the exercises.

#### 2.4.5. Electroencephalography (EEG)

EEG provides a set of observations (data), building a time series about the electrical activity of the brain. So, for this assessment a good level of EEG quality must be provided. Data will be collected at the frequency range of 0.16 to 43 Hz to categorize them in delta (<4 Hz), theta (4–7 Hz), alpha (7–13 Hz), and beta (14–30 Hz) waves associated with the duration and brain area of each type. These data can be presented as a heat map. To guarantee data quality, a minimum level of 75% connectivity will be set up prior to the experiment. A 32-channel Emotiv EPOC FLEX (EMOTIV PRO, 2022) will be used in EEG, which combines the flexibility and reliability of a capped electroencephalograph with wireless functionality and quick setup (20–30 min). The cortical delta, theta, alpha, and beta waves, locations, amplitude, and frequency will be collected in periods of 150 s in each exercise/side. After, amplification and digitization data are sent to a computer or mobile device for storage and data processing. Additional software will be used to read, interpret, and analyze brain waves (MatLab/MathWorks, 2021, Natick, MA, USA). These procedures allow for the identification of patterns and enrich the analysis performed [50].

#### 2.4.6. Video Motion

A Qualisys Video Motion System (Qualisys Medical AB, Sweden) equipped with 10 Oqus high-speed cameras will be used to analyze the quality of movement. This system performs real-time 3D video motion analysis, providing clear images of human movements, which allows for the analysis of trunk stabilization during exercise. The cameras are fixed on the laboratory walls, providing a complete view of the Standing Machine. The frequency used was 100 Hz. Calibration procedures were performed daily, according to the equipment manual. Forty-five reflective markers will be placed in anatomical landmarks (Figure 3) by two trained and experienced researchers [51,52].

A final anatomical check of markers will be carried out by a third researcher. The anatomical landmarks and markers were defined based on a literature review and in consultation with the operator, as shown in Table 4.

The quality of the movement for two exercises will be analyzed after this. The performance of the groups will be differentiated better in the time dimension by classifying them on a 5-point Likert scale. The scale ranges from 1 to 5, with 1 being labeled as very poor and 5 as very good performance.

#### 2.4.7. Qualitative Research

The process of combining collected data with both quantitative and qualitative methods will significantly enhance our understanding and provide insights into the phenomenon associated with the task. This approach allows us not only to explain but also to gain a deeper comprehension of the task. According to Creswell’s classification (2007) [53], the present study is categorized as a mixed study of the embedded type, due to the use of both qualitative and quantitative methods to collect data, which were carried out concurrently but with unequal weight. To proceed with this treatment, a content analysis will be carried out using NVivo software (QSR International, 2020, Burlington, MA, USA), Version number (Nvivo 14) with the definition of a priori and a posteriori categories. The interview guide will be constructed through expert submission, suggestion review, pre-testing, and application.

The interviews will be carried out by two researchers following pre-established guidelines. The interviews are expected to be recorded, typically lasting between 3 to 5 min. Subsequently, the recorded interviews will be processed. The researchers will also document the key responses from the participants in an interview grid. This comprehensive approach is intended to enhance the study’s credibility, transferability, dependability, and confirmability [54,55].

### 2.5. Operationalization/Sequence

Data will be collected after a standardized warm-up routine. After signing the informed consent forms, participants will be equipped with all technical devices needed in the “setup area”. Then, in the “calibration area”, the devices will be equipped, and the eye-tracker and EEG will be calibrated. Sequentially, an explanation of the Standing Plank exercise on a sliding platform will be presented. To ensure consistency in both content and tone, a pre-recorded instructional video will be used. Following the completion of the standing plank exercise, a second video demonstrating the side crisscross exercise will be shown. Before commencing the exercises, a 30-s setup will be recorded. Complete failure of one or both exercises will be allowed. After the data collection is completed, the subject proceeds to the “set-up area” for material disassembly procedures.

When the participants finish data collection, they will be moved to another room where the questionnaire on body image satisfaction will be completed and a semi-guided interview on the feelings and sensations of the practice will be conducted.

### 2.6. Statistical Analyses

This study seeks to contribute to the explanation and understanding of phenomena related to Pilates using a mixed quantitative and qualitative methodology. For the statistical treatment and linear analysis, IBM SPSS Statistics 28.0 software (IBM Corporation, New York, NY, USA) will be used, with a level of significance at 5% (*p* < 0.05). Descriptive statistics, including the calculation and analysis of the mean and standard deviation, will be conducted for all variables. If validation of the normality and heterogeneity assumptions of the sample are assured, then the analysis of variance ANOVA [56] will be used with multiple comparisons of means performed using the post-hoc Tukey test to verify the differences between the groups [57]. The effect size (ES) for ANOVA test is interpreted using the following criteria: no effect (ES < 0.04), minimum effect (0.04 ≤ ES < 0.25), moderate effect (0.25 ≤ ES < 0.64), and strong effect (ES ≥ 0.64) [58]. Apart from the effect size, the power (π) of the corresponding test will also be presented [59]. Pearson’s correlation analysis is defined to establish the associations between the studied variables [60].

While linear measures are useful for quantitative data analysis, taking a different perspective, non-linear data analysis is employed to characterize the collected data. This type of analysis provides insights into the organization of data within a system, offering a more nuanced understanding of the underlying patterns and structures in the dataset. The regularity, variability and complexity of the signal and the motor system in different levels of response [61,62] can lead to a larger understanding of the collected data. In fact, sample entropy can tell the amount of uncertainty attending the order of an output signal, i.e., predictability [63]. On the other hand, the Lyapunov exponent will be used to analyze the periodicity and regularity of the signal and the motor system in a time series [61]. Finally, the Hurst exponent [64,65] will be used to measure the tendency of the values of the motor system along a time scale. To proceed with the computation of the time series, the program will use UPATO [66].

### 2.7. Timetable

A schedule of all the tasks included in this project is presented in Table 5.

## 3. Results

In general, we anticipate both intra-individual and inter-individual differences that may result in statistically significant variations in the studied variables. This expectation considers the sample’s considerable diversity and variability in terms of morphological characteristics and psychomotor profiles.

Given that we have adopted both linear and non-linear analyses, which will provide a more comprehensive assessment of how the sample has evolved in terms of neuromuscular structures over both space and time, it is expected that the results may exhibit both quantitative and qualitative aspects. This will enable a mixed and more robust analysis of the applied methodology. For instance, within the context of non-linear analysis, we expect that results will show higher entropy in the expert groups when compared to non-experts. Regarding stability, the expert group is expected to exhibit higher Lyapunov exponent values compared to other groups, accompanied by lower Hurst exponent values. In the context of elastography, we anticipate that the elasticity of the transversus abdominis (TrA) muscle will be higher in experienced practitioners compared to those with less experience, both within the same group over time and when comparing across groups.

In terms of gaze behavior, results should reveal a greater number of saccades, more Areas of Interest (AOIs), and less time spent in fixation for the non-expert sample. This is because expert practitioners are assumed to have a more efficient mastery of gaze behavior.

## 4. Conclusions

Empirical research on Pilates practice has predominantly centered around showing its positive outcomes and advantages. This protocol seeks to introduce comprehensive models of these outcomes, expanding the observational perspective of this method. The significance of the methodological choices made is highlighted by the noticeable gap in the availability of data in Pilates-related studies. The introduction of relatively innovative techniques and analyses into Pilates-based exercises will allow us to begin contributing to the migration of practices developed in other areas to sports science. Achieving the most reliable knowledge will fulfil the needs of Pilates studies, adding to research on the learning process, motor control, and neuromuscular stabilization of the trunk, according to different expertise levels.

## 5. Patents

The Standing Machine has been submitted to patent pending number No 118908 in National Institute of Industrial Property.

## Figures and Tables

**Figure 1 healthcare-12-00229-f001:**
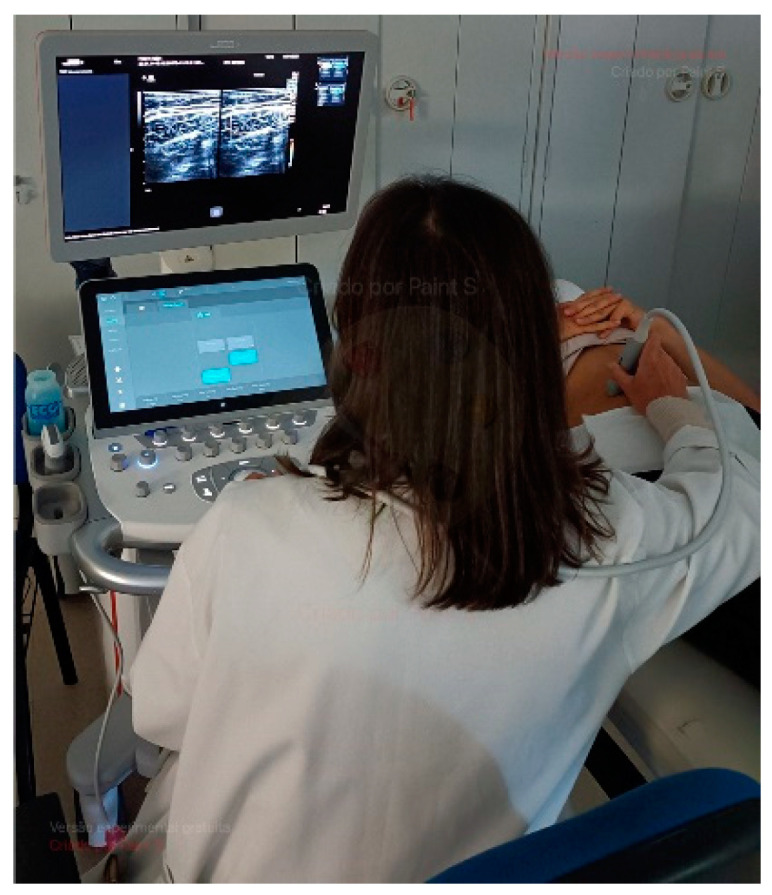
Elastography ultrasound (the image does not require a copyright).

**Figure 2 healthcare-12-00229-f002:**
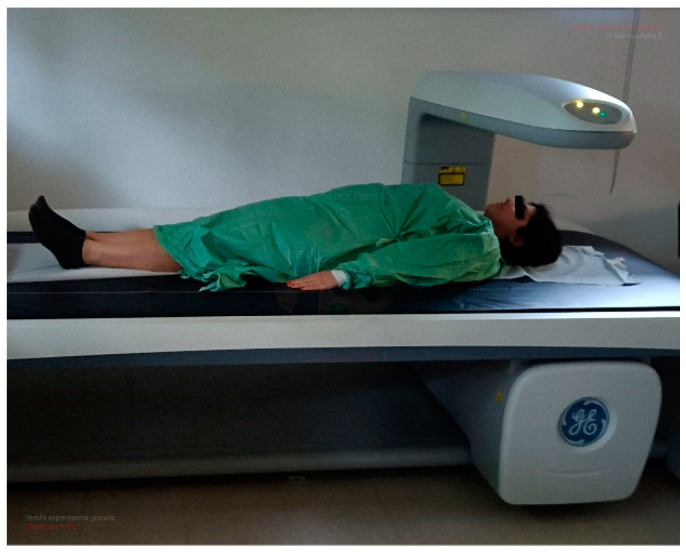
Example of DEXA exam (the image does not require a copyright).

**Figure 3 healthcare-12-00229-f003:**
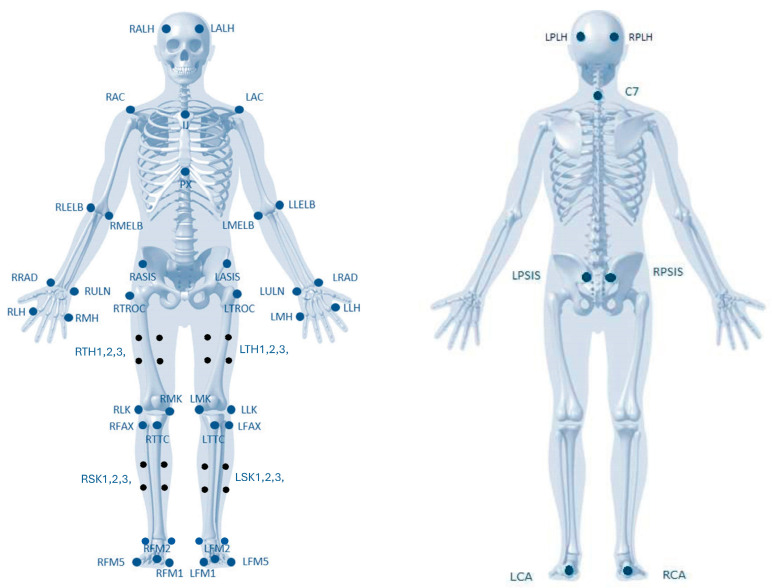
Figure of marker set placement (adapted from ROBOCORP Laboratory; the image does not require a copyright).

**Table 1 healthcare-12-00229-t001:** Study variables.

Dependent Variables							
			AWMUS + SWE	EEG	GAZE	QUALI	Video *
Independent Variables	Level of experience in Pilates	Control group	S1	S2	S3	S4	-
		Non-expertise group					
		Expertise group					
	Mirror	Mirror image	S1	S2	-	S4	-
		No mirror image					
	DEXA *		-	-	-	-	-

Legend: S1—study 1; S2—study 2; S3—study 1; S4—study 4; ECO—Abdominal Wall Muscle Ultrasound (AWMUS); SWE—Shear Wave Elastography; EEG—Electroencephalography; GAZE—Gaze behavior Assessment; QUALI—Qualitative research; DEXA—Dual-energy X-ray Absorptiometry. * Correlated variables—Used to establish analysis and identification correlations with other variables.

**Table 2 healthcare-12-00229-t002:** Sample operationalization and descriptive data.

Topic	Description	Operationalization	
Age	Age range between 30 and 50 years old		No pathologies that make physical practice unfeasible (according to RGPD and DL nº5/2007)
Gender	Gender Balance (approximately 20 males and 20 females)		
Experience level	Expertise	Practitioners who attend Pilates sessions in one of its forms, for more than 6 months with a frequency equal to or greater than two weekly workouts lasting between 45 and 60 min [27]	
	Non-expert	Practitioners who have never performed Pilates sessions in any of its forms or subjects who have performed Pilates for less than 3 months for more than a year will also be admitted to this group [27].	
Health condition		Unhealthy subjects will not participate in the study according to Portuguese law. The following will exclude participants from the sample: glaucoma, epilepsy, or pregnancy [27]	

**Table 3 healthcare-12-00229-t003:** Markers of anatomical assessment for each muscle [13,31].

Muscle	Anatomical Marker
RA	Measurements on the right side only Probe: Vertical Zone: 10 cm to the left of the umbilical scar at the midpoint between the iliac crest and the last rib
EO	
OI	
TrA	Measurements on the right side only Probe: Horizontal Zone: At the level of the navel and 2 cm to the right of the mid-axillary line

**Table 4 healthcare-12-00229-t004:** Markers of anatomical assessment for each muscle.

Marker Name	Location
RALH	Approximately over the temple and preferably aligned with the lateral commissure of the eye
LALH	
RPLH	Over the Occipital bone and at the same level as RALH and LALH on the frontal and sagittal plane
LPLH	
RAC	Acromial edge of the scapula
LAC	
C7	7th Cervical Vertebrae
IJ	Jugular Insertion/Notch of the Sternum
PX	Xiphoid Process of the Sternum
RLELB	Lateral Epicondyle of the Humerus
LLELB	
RMELB	Medial Epicondyle of the Humerus
LMELB	
RRAD	Radio-Styloid Process
LRAD	
RULN	Ulna-Styloid Process
LULN	
RLH	Lateral portion of the 5th metatarsal head
LLH	
RMH	Medial portion of the 5th metatarsal head
LMH	
RASIS	Anterior Superior Iliac Spine
LASIS	
RPSIS	Posterior Superior Iliac Spine
LPSIS	
RTROC	Trochanter
LTROC	
RLK	Lateral Condyle of the Femur
LLK	
RMK	Medial Condyle of the Femur
LMK	
RFAX	Proximal tip of the head of the Fibula
LFAX	
RTTC	Most anterior portion of the Tibia Tuberosity
LTTC	
RLA	Lateral prominence of the lateral Malleolus
LLA	
RMA	Medial prominence of the lateral Malleolus
LMA	
RCA	Distal end of the posterior aspect of the Calcaneus. Should be vertically aligned with FM2.
LCA	
RFM1	Lateral aspect of the 1st metatarsal head
LFM1	
RFM2	Dorsal aspect of the 2nd metatarsal head.
LFM2	Calcaneus marker should be vertically aligned.
RFM5	Lateral aspect of the 5th metatarsal head
LFM5	

**Table 5 healthcare-12-00229-t005:** Timetable for each task included in the project.

Stage	Estimated Dates	Interviewers	Output
Project submission to the Ethics Committee (EC)	September 2022	PhD student	Project and fill out the EC form
Pilot experiment	February 2023	PhD student/small sample/18 volunteers	Published 1 study (pilot experiment)
Presentation of the Project to the Scientific Council (CC) of FCDEF	October 2023	PhD student	Project
Experiment	September 2024	PhD student/large sample/27 volunteers	1–2 published articles

## Data Availability

The authors will share their research data after the study occurs.

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
