# Peer review of "Methodology and Experimental Protocol for Studying Learning and Motor Control in Neuromuscular Structures in Pilates"

_healthcare, 2024, doi:10.3390/healthcare12020229_

Round 1

Reviewer 1 Report

Comments and Suggestions for Authors

The study is well written.  It is explained nicely and it is easy to follow. 

I do recommend that on line 118 you do not mention the city and country of the location of the study.  For example, you might want to write, “The sample for the experienced group will be selected from a local Pilate studio.”

Also, the conclusion needs more to it.  It comes across as brief and you and the other authors can add more.  Write up how important this information can be for the general public and in the schools.

Reviewer 2 Report

Comments and Suggestions for Authors

The authors proposed a protocol for studying learning and motor control in neuromuscular structures in Pilate.  I have problems as the following:

1.The Table 1 needs to be reorganized for reading. 

2. I wonder whether other factors, such as BMI, might affect those dependent variables in Table 1.  As shown in Table 2, the authors only seemed to control the match of age and gender, what is the reason of these settings? Please justify.

3. The authors should clarify the sample size calculation process. 

4. I wonder why only the right side is included in Table 4? Will the dominant hand of participants be controlled consistently?

Reviewer 3 Report

Comments and Suggestions for Authors

Pilates is widespread but notoriously difficult to study well, so the researchers are to be commended for planning this work. This planned research on the Pilates physical conditioning method has an independent variable (Learner Expertise in Pilates) with two levels (Expert, Non-expert) and six dependent variables (Muscle Ultrasound, Shear Wave Elastography, Gaze Behavior, Electroencephalography and Video Motion Assessment).

While there are 6 pages describing the six outcome measures, there is only a single paragraph on the independent variable. However, inferences about the outcome of research are reliant on the definitional clarity of the levels of the independent variable, as well as their influence on the associated issue of internal validity (Campbell & Stanley, 1966, Experimental Designs for Research).

Here the participants are to be instructed about performing two Pilates-based skills – the Standing Plank and the Side Criss-Cross. Instructors are to have their training from a trade-marked, privately-owned organization, POLESTAR. The methods, criteria, standards and scoring used by this group to train Pilates instructors are not stated. One problem for the archival scientific literature with this approach is that when training companies are sold, the new owners may change methods, criteria etc. to their own preferences, and very soon the new company bears little resemblance to the old.

Because published science needs sufficient detail for replication, more than a brand name is required for future researchers to be able to repeat a study. More detail on the current POLESTAR training method for Pilates instructors is therefore needed. Even doing their training in different years may have already resulted in differences in instructors due to a shift in emphasis in POLESTAR methods over time. A sample of instructors encountered in different Pilates studios will show some focused on form, others on speedy performance of activities, and others who emphasise breathing, where all of them claim that their approach is ‘true Pilates’. Because Josef Pilates died in 1968, there is no way of confirming this. Things have moved on since ‘Contrology’.  Also, some studios favour the reformer, some the chair, and some favour mat work. So, the only solution here is to provide adequate detail about the relevant factors in the training of the instructors, without reproducing work that breaches copyright, so that later researchers using instructors trained by a different company can replicate this research.

Next, the entry criteria for participants into the two levels need attention. Presumably the researchers want to make inferences about any significant difference between the groups on the measures of interest that relate to their Pilates experience. However, if the groups differ significantly on anthropometric variables (eg. BMI, height, age) it may be argued that these factors caused the observed differences.

Arguably, it would be better to try to match entrants to the two groups on anthropometric variables, and score the amount of Pilates experience potential participants had received, then sort them into High and Low Pilates experience.  Trainers should be blind to the Experience level of the person they are training, so they do not subtly vary their teaching technique. Performance on the two Pilates-based skills should also be scored, using something like a 5 point Likert scale from Very Poor up to Very Good performance, so there can be pre-post evidence of acquisition. Complete failure to acquire the Criss-Cross skill, which looks difficult, should be contemplated.

Round 2

Reviewer 2 Report

Comments and Suggestions for Authors

The authors have revised the manuscript accordingly, and I am satisfied how the authors fulfilled previous requirements. I am suggesting manuscript acceptance.